# AI reveals insights into link between CD33 and cognitive impairment in Alzheimer's Disease

Tamara Raschka[1,2,3]*, Meemansa Sood[1,2], Bruce Schultz[1], Aybuge Altay[1,2¤], Christian Ebeling[1], Holger Fröhlich[1,2]*

**1** Department of Bioinformatics, Fraunhofer Institute for Algorithms and Scientific Computing (SCAI), Sankt Augustin, Germany, **2** Bonn-Aachen International Center for Information Technology (B-IT), University of Bonn, Bonn, Germany, **3** Fraunhofer Center for Machine Learning, Sankt Augustin, Germany

¤ Current address: Department of Computational Molecular Biology, Max Planck Institute for Molecular Genetics, Berlin, Germany

* tamara.raschka@scai.fraunhofer.de (TR); holger.froehlich@scai.fraunhofer.de (HF)

**Data Availability Statement:** The source code used to produce the results and analyses

## Abstract

Modeling biological mechanisms is a key for disease understanding and drug-target identification. However, formulating quantitative models in the field of Alzheimer's Disease is challenged by a lack of detailed knowledge of relevant biochemical processes. Additionally, fitting differential equation systems usually requires time resolved data and the possibility to perform intervention experiments, which is difficult in neurological disorders. This work addresses these challenges by employing the recently published Variational Autoencoder Modular Bayesian Networks (VAMBN) method, which we here trained on combined clinical and patient level gene expression data while incorporating a disease focused knowledge graph. Our approach, called iVAMBN, resulted in a quantitative model that allowed us to simulate a down-expression of the putative drug target CD33, including potential impact on cognitive impairment and brain pathophysiology. Experimental validation demonstrated a high overlap of molecular mechanism predicted to be altered by CD33 perturbation with cell line data. Altogether, our modeling approach may help to select promising drug targets.

## Author summary

For the last 20 years, the field of Alzheimer's Disease (AD) is marked by a series of continuous failures to deliver demonstrably effective medications to patients. One of the reasons for the continuous failure of trials in AD is the lack of understanding of how targeting a certain molecule would affect cognitive impairment in humans. One way to address this issue is the development of quantitative system level models connecting the molecular level with the phenotype. In this paper we propose a novel hybrid Artificial Intelligence (AI) approach, named Integrative Variational Autoencoder Modular Bayesian Networks (iVAMBN), combining clinical and patient level gene expression data while incorporating a disease focused knowledge graph. The model showed connections between various biological mechanisms playing a role in AD and allowed us to simulate a down-expression of the putative drug target CD33. Results showed a significantly increased cognition and predicted perturbation of several biological mechanisms. We experimentally validated these

presented in this manuscript are available on a GitHub repository at https://github.com/traschka/iVAMBN. The ROSMAP and Mayo data is available at https://adknowledgeportal.synapse.org/Explore/Studies/DetailsPage?Study=syn21241740. The CD33 KO cell line data is available at: https://www.ncbi.nlm.nih.gov/geo/query/acc.cgi?acc=GSE155567.

**Funding:** This project has received partial funding from the Innovative Medicines Initiative 2 Joint Undertaking under grant agreement No 115976. This Joint Undertaking receives support from the European Union's Horizon 2020 research and innovation programme and EFPIA. This work was partially developed in the Fraunhofer Cluster of Excellence "Cognitive Internet Technologies" and partially supported via the Fraunhofer Center for Machine Learning. The funders had no role in study design, data collection and analysis, decision to publish, or preparation of the manuscript.

**Competing interests:** The authors have declared that no competing interests exist.

predictions using gene expression data from a knock-out THP-1 monocyte cell line, which confirmed our model predictions up to a very high extent. To our knowledge, we thus developed the first experimentally validated, quantitative, multi-scale model connecting molecular mechanisms with clinical outcomes in the AD field.

## Introduction

Alzheimer's Disease (AD) is a neurodegenerative disorder affecting about 50 million people worldwide, resulting in the inability to perform necessary, daily activities before leading to an often early death [1]. Despite decades of research and more than 2000 clinical studies listed on ClinicalTrials.gov, to date there is no cure, and all existing treatments are purely symptomatic [1]. New disease modifying treatments are urgently needed, but require a better mechanistic understanding of the disease.

A common starting point in this context is to map out the existing knowledge landscape about the disease. In the past few decades, a large number of databases have been developed in the bioinformatics community, such as databases for biological pathways (like KEGG [2], PathwayCommons [3], WikiPathways [4], Reactome [5]), drug-target interactions (like OpenTargets [6], Therapeutic Targets Database [7]), disease-gene associations (like DisGeNET [8]) or protein-protein interactions (like STRING [9], IntAct [10]). All these databases simplify the usage of the respective knowledge for algorithms and models, especially in the field of drug target identification. Moreover, none of these databases have been compiled in a disease focused manner. The Biological Expression Language (BEL) provides this opportunity and can be used to represent literature-derived, disease focused knowledge in the form of attributed graphs in a precise manner. For AD a knowledge graph has been published in [11] and represents the manually curated, disease focused mechanistic interplay between genetic variants, proteins, biological processes and pathways described in the literature, enabling the user to computationally query and integrate knowledge graphs into drug target identification algorithms.

One of the interesting molecules in the AD field is CD33, a transmembrane receptor protein expressed primarily in myeloid lineage cells. It has been associated with decreased risk of AD in GWAS studies [12–18] and discussed as a potential therapeutic target, for example via immunotherapy [14]. In an AD mouse model, a knockout of CD33 mitigated amyloid-$\beta$ clearance and improved cognition [13, 17, 18]. Similarly, a positive effect on amyloid-$\beta$ phagocytosis could be observed in CD33 knock-out THP-1 macrophages [16]. In humans a correlation between CD33, cognition and amyloid clearance is known, however, the concrete underlying mechanisms are still not well understood. There is an ongoing clinical trial that is testing the effects of a CD33 inhibitor in patients with mild to moderate AD (NCT03822208). Along those lines, the EU-wide PHAGO project (https://www.phago.eu) funded via the Innovative Medicines Initiatives aimed to develop tools and methods to study the functioning of CD33 and related pathways in AD in order to facilitate decisions about potential drug development programs.

While graphs are useful for describing the disease focused knowledge landscape about AD, the principal incompleteness of disease focused biological knowledge may result in disagreements to observed data. Moreover, graphs do not allow to produce quantitative insights and predictions. For this purpose ordinary (ODEs) and partial differential equations (PDEs) are frequently used in systems biology and systems medicine, as they are able to describe biological mechanisms in a quantitative way. However, their formulation requires a detailed understanding of biochemical reactions, which in the AD field is only available for specific processes, like

for example amyloid-$\beta$ aggregation [19, 20]. Moreover, fitting differential equations usually requires time resolved data and the possibility to perform intervention experiments (as knockdowns or stimulation), which is challenged by the fact that cell lines and mouse models in the AD field can most likely only mimic specific aspects of the human disease [21–23].

A principle alternative to differential equation systems are probabilistic graphical models and in particular Bayesian Networks (BNs), which are quantitative as well. However, standard BN implementations require normally or multinomially distributed data, which is not the case in many applications. Furthermore, structure learning of BNs is an NP hard problem, where the number of possible network structures grows super-exponentially with the number of nodes in the network [24]. Hence, modeling higher dimensional data with a BN raises severe concerns regarding structure identifiability.

Altogether, these challenges lead to the fact that the AD field lacks a comprehensive quantitative model of the interplay between relevant molecules and biological processes, including the role of CD33, up to the phenotype level.

In this work, we developed a—to our knowledge—first quantitative, multi-scale model focused on the multitude of mechanisms governing the CD33 molecule. Our model spans a variety of modalities, including gene expression, brain pathophysiology, demographic information and cognition scores. To address the challenges mentioned before, we started with a disease focused knowledge graph reconstruction, which we clustered into modules to significantly reduce dimensionality. In the following we use the term "module" to denote a set of objects grouped together. Subsequently, we relied on our recently published Variational Autoencoder Modular Bayesian Network (VAMBN) algorithm [25], which is a hybrid Artificial Intelligence (AI) approach combining variational autoencoders [26] with modular Bayesian Networks [27], which is able to model arbitrary statistical distributions. We trained VAMBN on joint clinical and patient level gene expression data while employing a clustered knowledge graph reflecting incomplete prior knowledge about disease mechanisms and their interplay. A simulated knock-down of CD33 and predicted downstream effects could be experimentally validated with gene expression data from a cell line. Overall, we believe that our work helps to move closer towards a systemic and quantitative understanding of the disease, which is the prerequisite for finding urgently needed novel therapeutic options.

## Results

In this work, we relied on AD patient data from the Religious Orders Study and Memory and Aging Project (ROSMAP) [28–30] for model training and specificity analysis and the Mayo RNAseq Study (Mayo) [31] for external validation and specificity analysis. The data was retrieved from the RNASeq Harmonization study through the AMP-AD Knowledge Portal. Table 1 shows an overview about the clinical characteristics of the AD patients, which were used for model training and external validation. These patient samples were selected, because for all of them gene expression data from post mortem cerebral cortex tissues was available. We would like to mention at this point that gene regulation and thus gene expression is tissue specific [32]. Available data of other brain regions, also of healthy controls were thus kept separate for specificity analysis. A more detailed description of the used samples in each step of the analysis can be found in S3 Note.

### Overview about modeling strategy

Fig 1 shows an overview about our modeling strategy, which we call integrative VAMBN (iVAMBN), combining clinical and patient-level gene expression data with disease focused knowledge graphs. The first step of our workflow compiles an AD focused knowledge graph

**Table 1. Patient statistics.** Shown are the number of patients, their age in years (with mean and standard deviation), sex, APOE genotype (binary encoding for at least one present E4 allele), MMSE score (with mean and sd) and Braak stage.

|  |  | ROSMAP | Mayo |
|---|---|---|---|
| no. patients |  | 221 | 82 |
|  | age | 87.95 ± 3.38 | 82.66 ± 7.61 |
| sex |  |  |  |
|  | male | 68 | 33 |
|  | female | 153 | 49 |
| APOE |  |  |  |
|  | 0 | 138 | 39 |
|  | 1 | 83 | 43 |
| MMSE |  | 13.16 ± 8.38 | - |
| Braak |  |  |  |
|  | 1 | 7 | - |
|  | 2 | 6 | - |
|  | 3 | 42 | - |
|  | 4 | 71 | 6 |
|  | 5 | 88 | 35 |
|  | 6 | 7 | 41 |

describing cause and effect relationships between biological processes, genes and pathologies. The generated graph consisted of 383 nodes and 607 edges. The graph was subsequently clustered into modules with the help of the Markov Clustering algorithm [33] to significantly reduce the number of variables for subsequent modeling steps. Markov Clustering was chosen

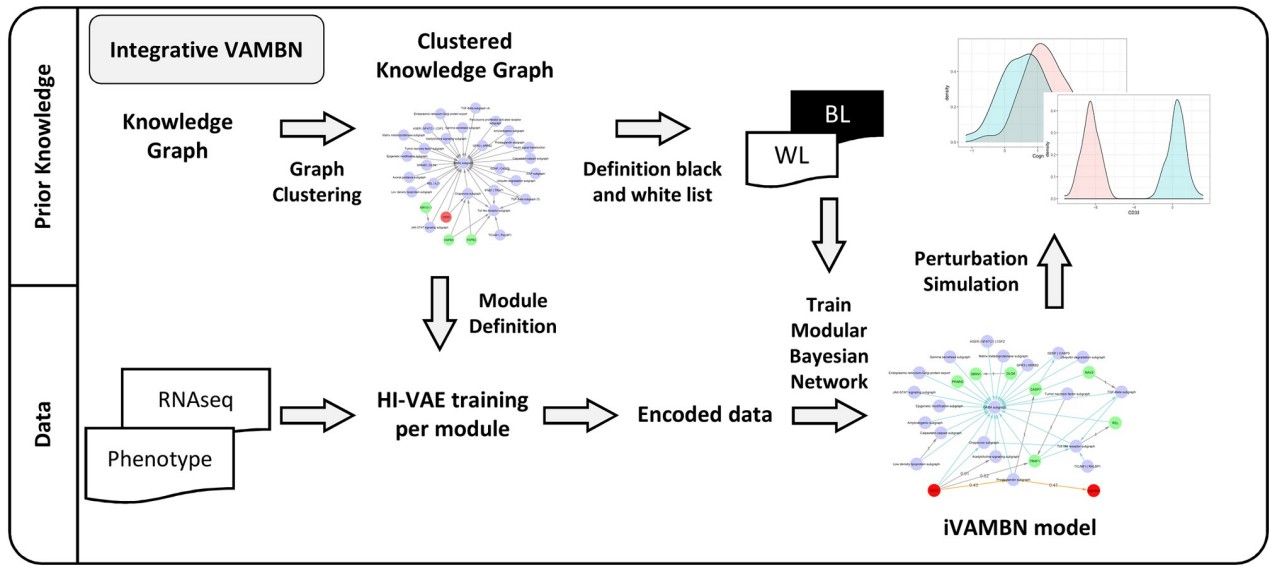

**Fig 1. The Integrative VAMBN (iVAMBN) approach.** The iVAMBN approach integrates gene expression data, clinical and patho-physiological (phenotype) measures (bottom left) into a joint quantitative, probabilistic graphical model. The method initially uses a knowledge graph (top left) for defining modules and for informing about potential connections between them. In a second step, a representation of each module using a Heterogeneous Incomplete Variational Autoencoder (HI-VAE) is learned. In a third step a modular Bayesian Network between autoencoded modules is learned while taking into account the information derived from the knowledge graph. Finally, the iVAMBN model is used to simulate gene perturbation (top right).

above other methods, as an evaluation of different graph clustering algorithms showed best metrics for this approach (see Methods). Genes within modules were annotated with AD disease mechanisms coming from the NeuroMMSig gene set collection [34].

Using patient-level clinical and gene expression data from post-mortem cerebral cortex tissues, in a second step the VAMBN algorithm was employed to quantitatively model relationships between gene modules as well as phenotype related scores (Mini-Mental State Examination (MMSE), Braak staging) and demographic features based on ROSMAP data. ROSMAP was chosen for training of the algorithm, because of the comparably large number of patients (more than 200) and available MMSE plus Braak scores. VAMBN takes as input patient-level data hierarchically organized into pre-defined modules (here: either gene modules or a phenotype related module including i.e. MMSE plus Braak stages), original features (here: demographic and clinical variables like age, sex, APOE genotype, and brain region) and prior knowledge regarding their possible connections. The output is a probabilistic graphical model describing connections between modules and original features. There is a per-patient score for each module, and each of these scores can be further decoded into feature-level gene expression and phenotype data, respectively.

In the third step of our strategy we evaluated, whether our iVAMBN model could also explain gene expression data from the Mayo study. Notably, at this step we only considered the Braak stage in the phenotype module, because the Mayo study does not report MMSE scores. For that purpose we first re-trained our iVAMBN model on ROSMAP while leaving out MMSE scores and then assessed the marginal log-likelihood of the modified model on the Mayo dataset. We then tested the marginal log-likelihood of the true model against randomly permuted versions of the learned probabilistic graph. This allowed us to assess, in how far the model learned on ROSMAP could explain Mayo data better than expected by pure chance.

For the last step, we used our iVAMBN model trained on ROSMAP to simulate several therapeutic interventions, including a CD33 inhibition. Based on available data, we were able to experimentally validate the predicted effects of a CD33 inhibition using CD33 knockout gene expression data from a THP-1 monocyte cell line. More details about the entire iVAMBN approach can be found in the Methods section of this paper.

In the following we elaborate on the results obtained in each of these different steps, while technical details are provided in the Methods part of this article.

## Knowledge graph compilation

As outlined in the previous section, our modeling approach started with the compilation and Markov clustering of a knowledge graph. The Markov clustering resulted in 32 modules, including 4 single gene modules, namely CD33, HSPB2, HSPB3, and MIR101–1. Most of the non-single gene modules comprised only two genes, while others had multiple combinations, like the GABA subgraph module with 289 genes. The exact number of genes clustered together as well as the result of a statistical over-representation analysis (hypergeometric test) using the AD focused gene set collection NeuroMMSig [34] can be found in S1 Table. A complete list of molecules within each module can be found in S2 Table. The modules were considered as nodes of a graph between them, where an edge was set between modules $M_1$, $M_2$, if in the original knowledge graph there was at least one gene in $M_1$ and one in $M_2$ that was connected via a directed path. The resulting (acyclic) module graph is shown in S1 Fig.

## Integrative variational autoencoder modular bayesian network model

Integrative VAMBN combines the advantages of Bayesian Networks with the capabilities of variational autoencoders, more specifically Heterogeneous Incomplete Variational Autoencoders

(HI-VAEs) [35]. Briefly, the idea is to initially learn a low dimensional Gaussian representation of features mapping to each of the defined modules. HI-VAEs differ from classical variational autoencoders in the sense that they can be applied to heterogeneous input data of different numerical scales, potentially containing missing values. In a second step a Bayesian Network structure is then learned over the low dimensional representations of modules, resulting in a modular Bayesian Network. More details are presented in the Methods part of this paper and in [25].

We here trained an iVAMBN model using the identified modules (i.e. feature groups in the original data) as—potentially multivariate—nodes of a probabilistic graphical model. Noteworthy exceptions are described in detail in S1 Note. In cases where multiple features map to one and the same module (i.e. the corresponding node / random variable in the probabilistic graphical model is multivariate), our method initially learns a low dimensional representation using a HI-VAE. Second, we learned the Bayesian Network structure connecting these modules. At this stage it is possible to provide information about possible connections between modules given in the knowledge derived module graph (S1 Fig). We tested three different strategies to incorporate the information provided in the module graph:

- *completely data driven*: the entire Bayesian Network was only learned from data,

- *knowledge informed*: the module graph was either used to only initialize Bayesian Network structure learning, to enforce / white list the existence of specific edges, or used for a combination of both, and

- *completely knowledge driven*: strictly constrain edges between modules to those provided via the module graph, and additionally learned ones are only allowed to connect cognition scores, patho-physiological stages, and demographic features. All other possible edges are black listed, i.e. not allowed.

A systematic comparison of these strategies via a cross-validation yielded a better performance of the second strategy (knowledge informed), in which we used the module graph to white list edges and to initialize a greedy hill climbing based structure learning, see details in Methods Section and S2 Note. That means, iVAMBN was allowed to add additional edges, if the data provided according evidence.

We repeated the knowledge informed modular Bayesian Network learning 1000 times on random bootstrap sub-samples of the data drawn with replacement, hence allowing to quantify the statistical confidence of each inferred edge. The results of this analysis can be found in S3 Table.

In the following we only focus on the 130 edges that were found in at least 40% of the 1000 modular Bayesian Network reconstructions (Fig 2). Notably, this threshold was only chosen for better visualization purposes and to limit the subsequent discussion. Edges with lower bootstrap probability might also exist in reality despite lower statistical confidence. Nodes corresponding to sex, APOE status, and brain region were not connected to any other nodes with sufficient statistical confidence, meaning that these features might have no impact on the rest of the network. Nodes with only outgoing edges in the network (i.e. source nodes) were: the years of education, the age, and the single gene NAV3. The GABA subgraph (containing more than 280 genes) and the phenotype module were leaf nodes, meaning they had no outgoing edges. Only patient age had a direct influence on CD33. CD33 had eight directly influenced molecular mechanisms: the GABA subgraph, the Amyloidogenic subgraph (containing genes SRC and APBA2), the Acetylcholine signaling subgraph (containing genes ACHE and PRNP), the Prostaglandin subgraph, and the Chaperone subgraph (containing genes HSPB6, CXCL8,

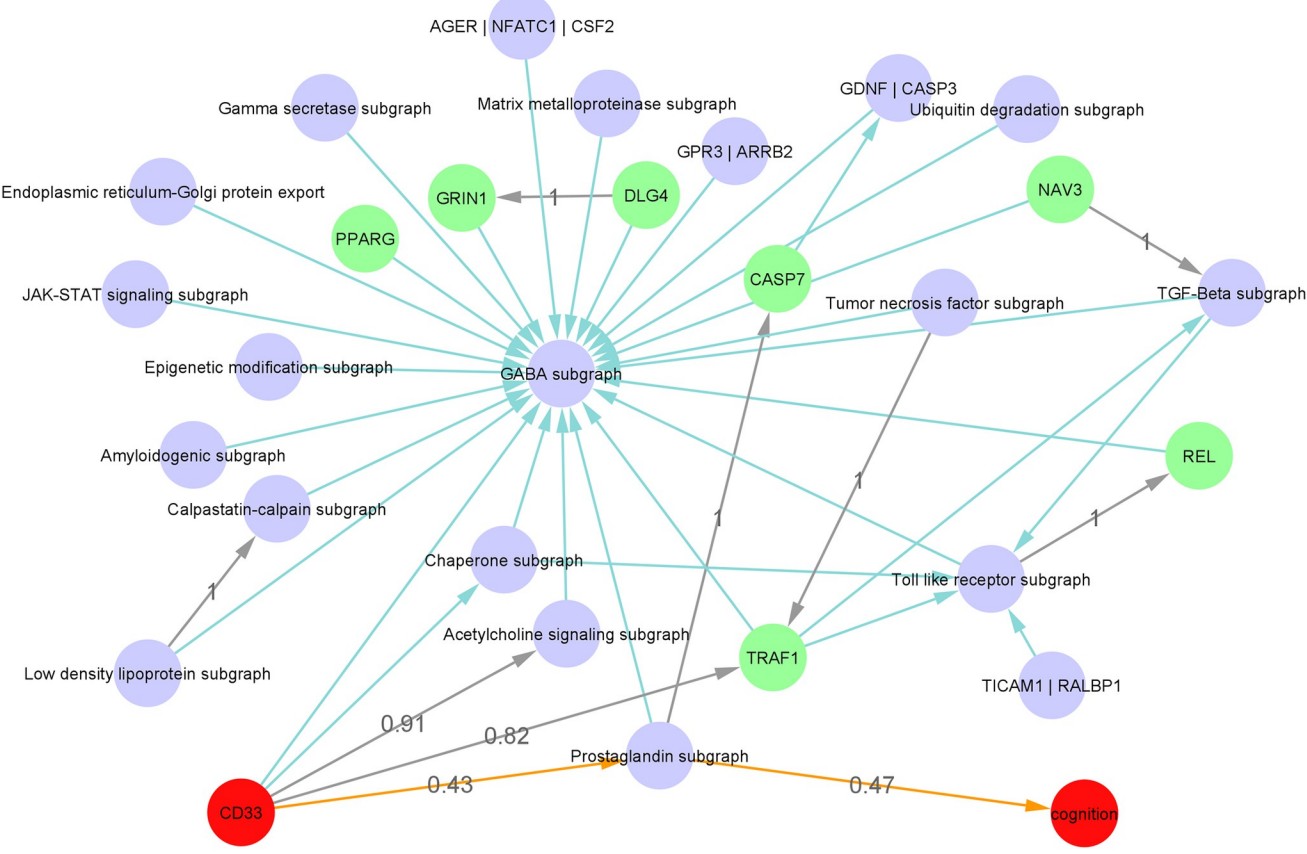

**Fig 2. Network representation of iVAMBN model for ROSMAP data.** Shown are the learned (grey) and knowledge-derived (green) edges between gene modules (purple nodes), single gene modules (green) and CD33 and phenotype module (red). All these edges appeared with bootstrap frequency > 0.4. The newly inferred shortest path between CD33 and phenotype is displayed in orange. Other edges with bootstrap frequency > 0.4 have been removed for visualization purposes, except for those six edges which were trained with a bootstrap confidence of 1.

and CCR2). Also, the single gene module, TRAF1, was a child of CD33. Altogether, CD33 had a predicted causal influence on every node, except for the source nodes.

**Model reveals path between CD33 and disease phenotype.**   As shown in Fig 2 the shortest path between CD33 and the disease phenotype was observed through the Prostaglandin subgraph. All the edges from this connection were newly learned from data, meaning that they had not previously been identified in the knowledge graph. Nevertheless, these correlations have been previously reported in the literature: Prostaglandines are eicosanoides, which were found to play a role in memory learning and neuroinflammation [36, 37]. A major producer is activated microglia, which itself is activated through amyloid-$\beta$ and produces inflammatory cytokines [38]. Currently, microglia and their effects on AD is a major focus of research [39, 40]. Also, PGD2, a prostaglandin mainly synthesized in neurons, was previously found to be upregulated in AD patients [41]. Pairwise correlation plots between the genes of the prostaglandin pathway and CD33 or phenotype can be found in S2 Fig.

In total, 130 of the 162 edges of the bootstrapped iVAMBN model were newly learned from the data and had not been previously identified within the literature derived knowledge graph. Out of these 130 edges, six edges had a bootstrap confidence of 100%, meaning that they were learned consistently from 1000 random sub-samples of the data. A list of these edges can be found in Table 2.

**Table 2. Consistently newly learned edges in iVAMBN model.** All edges were found in each of 1000 network reconstructions from randomly subsampled data.

| from | to |
|---|---|
| DLG4 | GRIN1 |
| Tumor necrosis factor subgraph | TRAF1 |
| Toll like receptor subgraph | REL |
| Low density lipoprotein subgraph | Calpastatin-calpain subgraph |
| Prostaglandin subgraph | CASP7 |
| NAV3 | TGF-Beta subgraph |

These high confidence edges demonstrated strong pairwise correlations between connected modules. NAV3, for example, had a strong negative correlation with MAVS, a member of the TGF-Beta subgraph module (Fig 3 left). In contrast to that SRSF10 and CREB1, members of the Low density lipoprotein subgraph and Calpastatin-calpain subgraph modules, were strongly positive correlated (Fig 3 right).

Although no direct correlation between NAV3 and MAVS is known, their effects are both linked to AD. NAV3, which is predominantly expressed in the nervous system, is increased in AD patients [42], while MAVS encodes a gene that is needed for the expression of beta interferon and thus contributes to antiviral innate immunity and may protect the cells from apoptosis [43]. Together with the strong negative correlation seen in the data, one can hypothesize that the increased level of NAV3 in AD leads to a decreased level of MAVS, which elevates apoptosis of the cells.

The strong positive correlation between SRSF10 and CREB1 linked the Low density lipoprotein (LDL) and Calpastatin-calpain subgraphs. LDL is a major APOE receptor, which is the strongest genetic factor for AD, where different alleles are either risk or protective alleles [44]. APOE is also linked to amyloid-$\beta$, whose production is increased with elevated activity of calpain due to the decreased levels of calpastatin. Calpastatin is also linked to synaptic dysfunction and to the tau pathology of AD [45, 46]. Tau is another protein that accumulates in the brains of AD patients. The exact underlying mechanisms here are still unknown, but regulatory mechanisms of calpain are highly influenced by Calcium ($Ca^{2+}$) influx and increased

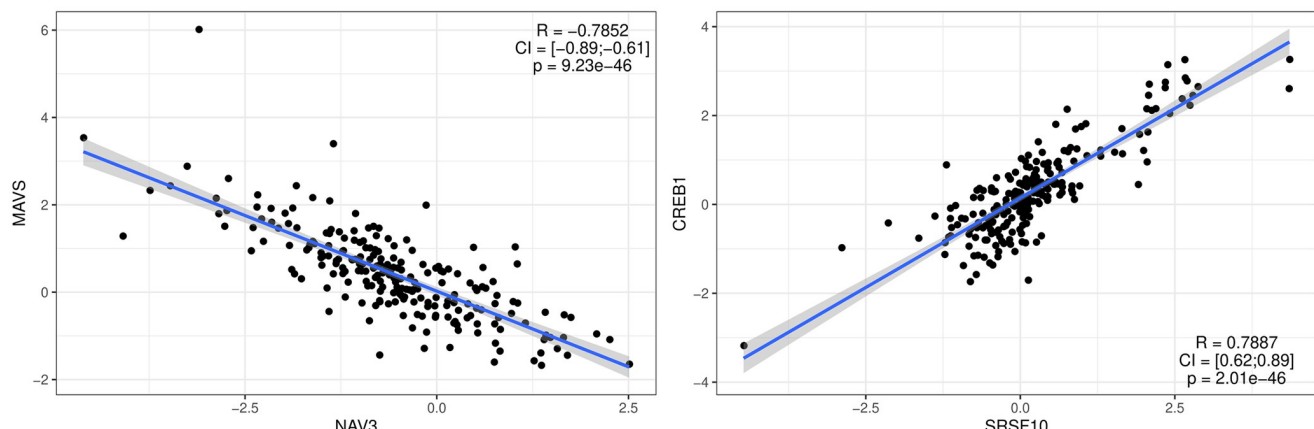

**Fig 3. Quantitative relationships learned by iVAMBN.** Each correlation (R) is shown along with its confidence interval (CI) and multiple testing adjusted p-value. Left: Correlation of NAV3 with TGF-Beta subgraph module member MAVS. Right: Correlation of Low density lipoprotein subgraph module member SRSF10 with CREB1, a member of the Calpastatin-calpain subgraph module. Further plots can be found in S2 and S3 Figs.

intracellular calcium levels are a main reason for the loss of neuronal function in AD [45–47]. Changes in the Calpastatin-calpain mechanism may therefore also lead to reduced amyloid-$\beta$ deposition.

### External validation of iVAMBN model

We assessed the ability of the model to explain normalized gene expression data from an independent study, Mayo. Notably, all gene expression data used in this analysis was mapable to the same brain region, namely the cerebral cortex, via the Uber-anatomy ontology (UBERON) [48]. However, Mayo does not contain MMSE scores. Therefore, we first trained a modified version of our iVAMBN model on ROSMAP, which only contained the Braak score in the phenotype module, but otherwise had the edges shown in Fig 2. The full list of edges of this model together with their corresponding bootstrap confidences can be found in S3 Table. We then explored the marginal log-likelihood log $p(data \mid graph)$ of the model on the Mayo dataset and subtracted the marginal log-likelihood obtained by 1000 random permutations of the network (Fig 4), resulting in an empirical p-value. This showed that our model could explain Mayo gene expression data significantly better than randomly permuted networks ($p = 0.035$) despite the clinical differences between patients in both studies shown in Table 1.

Moreover, we evaluated the ability of our iVAMBN to predict the activity score of the prostaglandin module in the Mayo study. The prostaglandin module was chosen, because prostaglandins play a role in neuroinflammation, which is a hallmark of the disease phenotype. Moreover, the prostaglandin module was directly connected to the clinical/pathological phenotype module, in which Braak stages, however, significantly differed between Mayo and

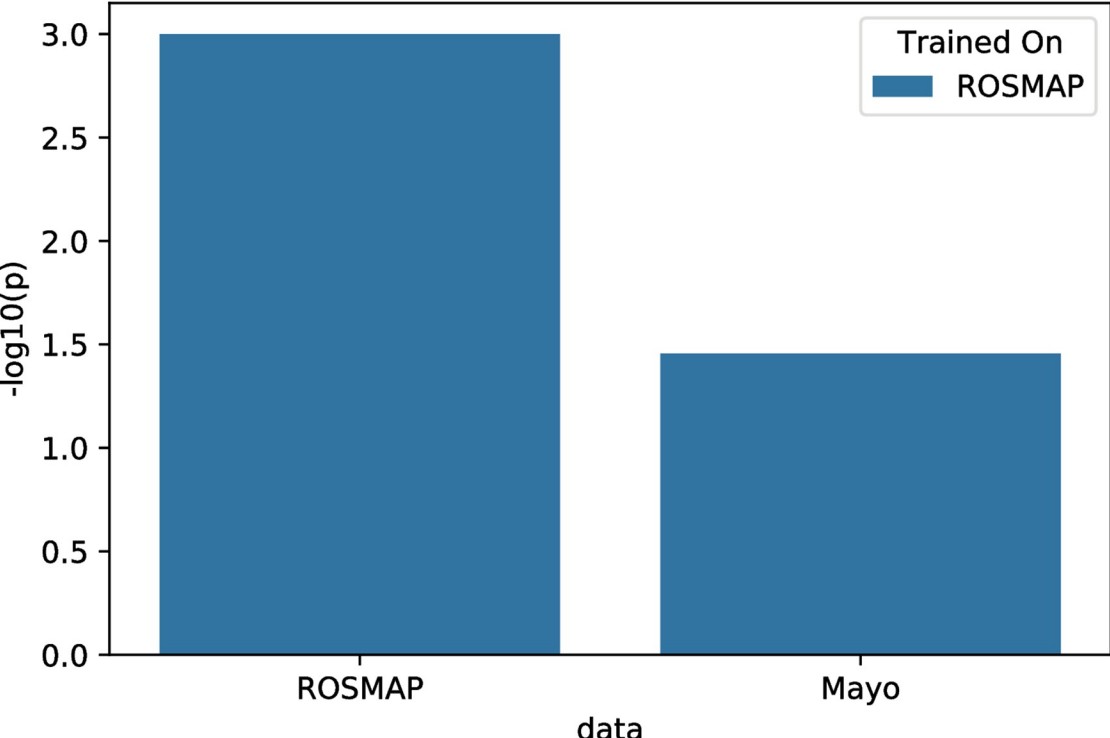

**Fig 4. External model validation.** Statistical significance $-log_{10}(p)$ value of the marginal log-likelihood of the model when evaluated on the training data (ROSMAP) and external validation data (Mayo).

ROSMAP studies. We thus regarded the activity score of the prostaglandin module as a relevant and sufficiently comparable surrogate endpoint between Mayo and ROSMAP studies. We used our iVAMBN model trained on ROSMAP to predict the activity score of prostaglandin module activity in Mayo by feeding data from genes outside the prostaglandin module to the model expression. We observed a highly significant Pearson correlation between true and predicted values in the external validation dataset ($r = 0.69$, 95% CI: [0.56;0.79]). Hence, we concluded that our iVAMBN model was predictive for the chosen endpoint.

Finally, we trained a separate iVAMBN model on Mayo data and explored the overlap with the ROSMAP model at different thresholds of the bootstrap confidence (S4 Fig). At the previously chosen 40% threshold the overlap of the newly learned edges contained in the iVAMBN models trained on ROSMAP and Mayo was statistically significant, even if edge directions were considered (hypergeometric test, $p < 1e - 38$).

## Specificity and sensitivity of iVAMBN model

**Specificity to brain region.**   We tested the ability of the model to explain normalized gene expression data from other brain regions. Therefore, we trained multiple additional iVAMBN models on patient samples belonging to the posterior cingulate cortex, the dorsolateral prefrontal cortex, and the head of caudate nucleus from the ROSMAP study, as well as on samples from the temporal cortex and the cerebellum from the Mayo study. We then investigated the overlap of each of these additional iVAMBN models with our primary one akin to the external validation described in the previous Section. Among non-cortical brain regions, the largest and statistically significant overlap on graph level was found with an iVAMBN model trained with samples from the head of caudate nucleus ($\sim 36\%$ considering edge directions). The lowest (still statistically significant) overlap was found with the cerebellum ($\sim 31\%$ considering edge directions). The primary iVAMBN model for all datasets was able to predict the activity score of the prostaglandin module, but the prediction performance was clearly lower in non-cortical brain regions (see results in Table B in S3 Note). Altogether our results thus suggest that our primary iVAMBN model is focused on cortical brain regions.

**Disease specificity.**   Similar analyses were done for iVAMBN models trained on available healthy control samples from the posterior cingulate cortex, the dorsolateral prefrontal cortex, and the head of caudate nucleus from the ROSMAP study. The graph structures still demonstrated a significant overlap with our primary iVAMBN model but were considerably lower, see Table C in S3 Note. This suggests that our primary iVAMBN model is AD focused.

**Sensitivity to knowledge graph.**   Finally, we explored, how sensitive our primary iVAMBN model was to the knowledge graph. For that purpose, we randomly shuffled all edges of the original knowledge graph, re-clustered this permuted graph, and re-trained a complete iVAMBN model. The iVAMBN model trained on the permuted graph demonstrated a significantly lower marginal log-likelihood $p(data \mid model)$ compared to the primary iVAMBN model ($p = 4.14E - 24$), see details in Supplements (Fig B in S3 Note). Hence, we concluded that our primary iVAMBN model was sensitive to the knowledge graph structure.

## CD33 down-expression simulation

To understand the potential systemic consequences of a therapeutic intervention into CD33 we used our primary iVAMBM model to simulate a down-regulation. This was achieved by a counterfactual down-expression (here: 9-fold) of CD33 in every patient (Fig 5 (top left)). Due to the fact that iVAMBN is a quantitative model, associated downstream consequences on biological mechanisms and phenotype could be predicted in every patient (see examples in Fig 5).

 AI reveals insights into link between CD33 and cognitive impairment in Alzheimer's Disease

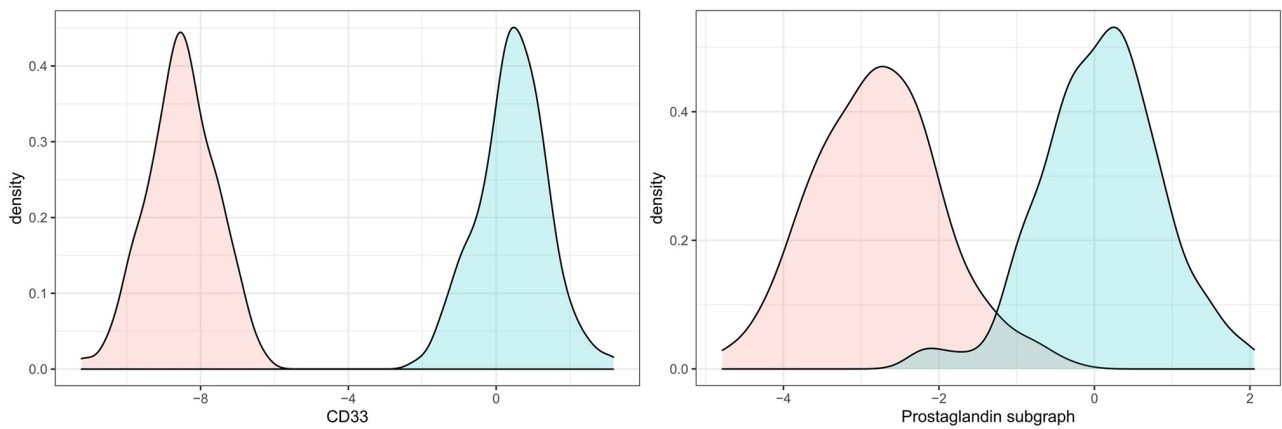

**Fig 5. Module distributions in original and simulated CD33 down-expression.** The blue curve describes the original distribution, while the red one describes the CD33 down-expression scenario. CD33 down-expression simulation (left) results in lower scores of the prostaglandin pathway module (right).

CD33 down-expression simulation (left) results in higher activity scores of the prostaglandin pathway module (right).

In addition, iVAMBN predicted a significant increase of MMSE scores ($p < 0.001$, Fig 6 (left)), and also a significant decrease of Braak stages ($p < 0.001$, Fig 6 (right)). That means patients are not only predicted to improve the specific cognitive abilities tested by MMSE, but are also predicted to improve brain pathophysiology.

**CD33 down-expression reveals significant changes in many mechanisms.** Our iVAMBN model predicted significant effects on gene expression of 28 mechanisms and individual genes, respectively (Table 3). Significant changes were, for example, predicted for the genes CASP7 and TRAF7, and the prostaglandin and calpastatin-calpain mechanisms. But also the amyloidogenic mechanism is significantly differential expressed in a CD33 knockdown scenario.

Decreased expression of the amyloidogenic mechanism will thus result in patients with less amyloid-$\beta$ deposition. While this connection of the amyloidogenic mechanism and AD is clear, others need to be further explored.

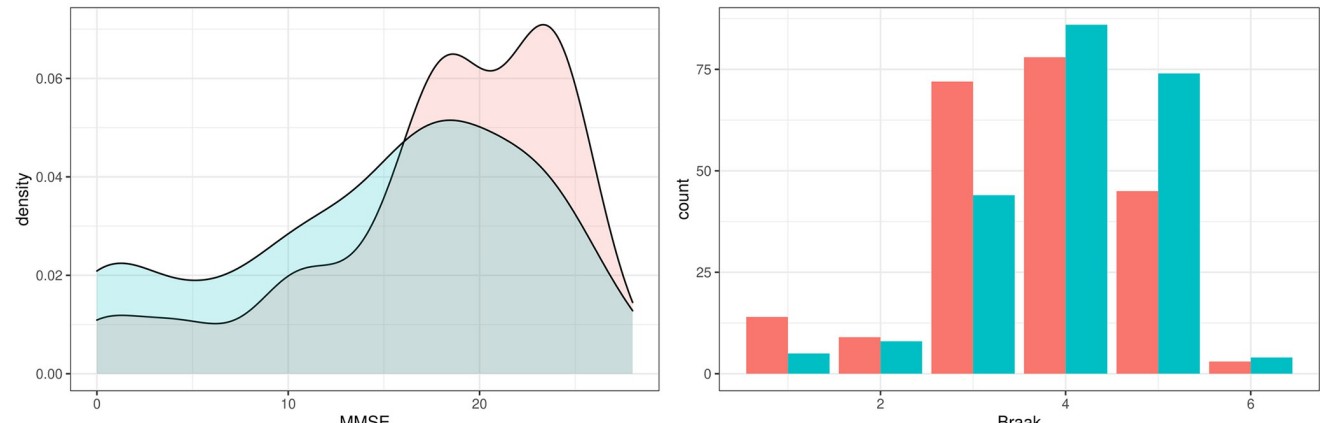

**Fig 6. Predicted changes on phenotype (MMSE and Braak stages) as a consequence of CD33 down-expression.** Distribution of MMSE and Braak stages in CD33 original (blue) and down-expressed (red) patients shows a significant improvement of scores and thus cognition as well as brain pathophysiology.

**Table 3. Statistical significance of gene modules.** The table shows results of a global test [49], assessing the differential gene set expression of each gene module between WT and down-expression/KO of CD33. P-values of the test within simulated scenario, as well as, p-values from cell line KO are reported and corrected for multiple testing using the Benjamini-Hochberg method. The agreement of both tests is described in the last column, meaning if both tests are either significant or non-significant (+) or if they don't show same direction of significance (-). For GRIN1 no p-value could be computed, as that gene is not present in the cell line data.

| Gene module | p-value simulated KD | p-value cell line KO | agreement significance |
|---|---|---|---|
| GABA subgraph | 2.75e-04 | 3.60e-15 | + |
| Toll like receptor subgraph | 1.05e-26 | 1.05e-13 | + |
| Prostaglandin subgraph | 6.99e-109 | 1.02e-09 | + |
| TGF-Beta subgraph | 0.592 | 8.79e-11 | - |
| Calpastatin-calpain subgraph | 3.14e-91 | 5.41e-09 | + |
| JAK-STAT signaling subgraph | 0.454 | 2.91e-11 | - |
| AGER / NFATC1 / CSF2 | 5.78e-41 | 0.0129 | + |
| Chaperone subgraph | 2.84e-75 | 2.02e-09 | + |
| REL | 4.45e-18 | 9.96e-11 | + |
| Ubiquitin degradation subgraph | 5.15e-20 | 1.06e-06 | + |
| GRIN1 | 1.92e-132 | NA | |
| PPARG | 2.20e-04 | 1.78e-03 | + |
| GDNF / CASP3 | 1.06e-17 | 2.98e-11 | + |
| Gamma secretase subgraph | 4.36e-10 | 1.93e-03 | + |
| Epigenetic modification subgraph | 6.90e-58 | 7.64e-03 | + |
| TICAM1 / RALBP1 | 1.46e-16 | 0.0561 | - |
| Amyloidogenic subgraph | 4.54e-69 | 9.11e-10 | + |
| Tumor necrosis factor subgraph | 0.0997 | 0.769 | + |
| Acetylcholine signaling subgraph | 6.74e-04 | 0.337 | - |
| Matrix metalloproteinase subgraph | 0.0708 | 2.74e-10 | - |
| NAV3 | 0.176 | 3.81e-07 | - |
| TRAF1 | 1.66e-95 | 2.26e-08 | + |
| CASP7 | 1.75e-138 | 0.151 | - |
| GPR3 / ARRB2 | 4.87e-04 | 8.02e-04 | + |
| Endoplasmic reticulum-Golgi protein export | 5.19e-29 | 1.78e-11 | + |
| Low density lipoprotein subgraph | 0.891 | 8.11e-06 | - |
| DLG4 | 5.85e-93 | 3.44e-07 | + |
| CD33 | 3.33e-307 | 8.06e-08 | + |

The link between Calpastatin-calpain mechanism and AD was already described earlier. The key aspect is its negative influence on amyloid-$\beta$ deposition. PGD2, a prostaglandin mainly synthesized in neurons, was previously found to be upregulated in AD patients [41]. Prostaglandines are eicosanoide, which were found to play a role in memory learning and neuroinflammation [36, 37]. A major producer is activated microglia, which itself is activated through amyloid-$\beta$ and produces inflammatory cytokines [38]. Currently, microglia and their effects on AD is a major focus in the field of research [39, 40]. Again, down-expression of the prostaglandin may result in reduced amyloid-$\beta$ deposition. Altogether, the vast majority of significantly differential expressed gene sets was highly linked to AD through the amyloid-$\beta$ cascade.

**Experimental validation with cell line data.** We checked whether our iVAMBN based predictions experimentally agreed with cell line gene expression data, specifically reflecting wild type (WT) and CD33 knock-out (KO). Our analysis (see details in Methods part) revealed significant changes of 23 AD associated mechanisms and genes in KO versus WT. Interestingly, 19 out of these 23 mechanisms overlapped with those predicted by iVAMBN (Table 3).

Likewise, iVAMBN predicted significant changes of 22 genes and gene sets, respectively, out of which only 3 were false positives at a false discovery rate threshold of 5%. Notably one of the false positive predictions (TICAM1 / RALBP1) had an adjusted p-value of 5.6% in the experimental data.

Overall, we thus observed a high degree of overlap between the dysregulated mechanisms and those predicted by the iVAMBN model, indicating that our model aligns well with the cell line data.

**Simulation of the perturbation of other candidate targets.**   For comparison reasons, we further simulated the effect on the phenotype of a 9-fold up- or down-regulation of all other genes in our model, which showed a directed path to the phenotype module. Genes belonging to modules which were not an ancestor of the phenotype module were excluded, because they could not have any effect on the phenotype according to our model. We simulated for each candidate target an up- as well as a down-regulation.

The simulated dys-regulations showed that none of the candidate targets had a predicted effect on the phenotype stronger than CD33 (S5 Fig). Only TRAF6 and TGFB3 down-regulation as well as up-regulation of APBA2, TRAF5 and SALL1 were predicted to increase the mean MMSE score by more than two points, compared to a predicted increase by almost five points via CD33 perturbation.

APBA2 is known to interact with APP and therefore plays a role in the amyloidogenic pathway [50, 51]. TRAF6 was identified in multiple experiments as target of miR-146a which is a key regulator of innate immunity that is up regulated in AD pathology affected brain regions and might also has an effect on amyloid-$\beta$ metabolism [52]. It was found that treatment with a miR-146a agomir inhibits TRAF6 expression and reduced the cognitive impairment in AD mice [53].

## Discussion

The here presented work is the first to demonstrate, to our knowledge, that one can integrate gene expression and clinical data together with qualitative knowledge about cause-and-effect relationships into a quantitative, system medical model of AD. This was achieved via an AI based method, which we combined with a knowledge graph representation of AD. We could show that a simulated CD33 down-expression agrees well with experimental gene expression KO data from a THP-1 cell line. Overall, our model could thus help to understand and quantify intervention effects on a multi-scale biological system level and thus aid the identification of novel therapeutic targets, which are urgently needed in the AD field.

Our model predicted that CD33 down-regulation would yield a significant effect on cognition (MMSE) and brain pathophysiology (Braak scores) through the prostaglandin pathway. Although the role of prostaglandins is known to play a role in memory, learning and neuroinflammation [36, 37], the exact mechanism by which cognition is affected remains unknown, but seems to be coupled to amyloid-$\beta$ deposition through microglia. In AD mice, a knockout of CD33 mitigated amyloid-$\beta$ clearance and improved cognition [17, 18]. A positive effect on amyloid-$\beta$ phagocytosis could also be observed in CD33 KO THP-1 macrophages [16].

Despite the evidence for a positive effect on cognition, we should mention that CD33 as a possible drug target has possible caveats that have been discussed in the literature [14]: i) It is not clear whether the genetic association of CD33 to AD is causal or just due to linkage disequilibrium with the true causal variant. ii) It is so far not entirely clear, how to therapeutically manipulate the expression level of CD33 in an optimal manner. iii) There might be safety issues due to the fact that CD33 is important for inhibiting immune responses and mediating self-tolerance. Systemic CD33 inhibition could potentially induce inflammatory autoimmune

diseases. We therefore see the investigation of CD33 conducted in this paper more as a showcase for our iVAMBN approach rather than making any specific recommendation regarding the therapeutic value of CD33. Integrating known side effects of approved drugs targeting specific proteins in our model's graph structure could provide hints on possible side effects and is an interesting point for further research.

Altogether we see the impact of our work two-fold: first, we have introduced a novel multiscale, quantitative modeling approach (iVAMBN), which is widely applicable in systems medicine, specifically in situations, where only a partial mechanistic understanding of biological phenomena is given. Secondly, our developed model can be further explored by the AD field and could aid a better understanding of the disease as well as identification of novel therapeutic options.

## Methods

### AD knowledge graph

A major part of this study is a BEL (https://bel.bio) encoded, knowledge graph, which was initially compiled via text mining and later on manually curated via literature. In general, the BEL language helps to build a computer-process-able cause-and-effect relationship model. Each BEL statement consists of a subject and an object, connected through a relation. Subjects and objects could be many different entities, like genes, proteins or RNA, but also biological processes, pathologies or even chemicals. Therefore, the relations have many different facets, as well. These could be relations like *increases*, *decreases* or *association*, describing the interaction between subject and object. But there are also relationships describing something like a membership of subject and object, for example *hasComponent* and *isA*. The BEL model used here, is an enriched version of the AD cause-and-effect relationship model published in [11] and can be found in the github repository. The enrichment was done around the two genes CD33 and TREM2, such that detailed knowledge about these two genes was gathered in the context of AD.

A filtering step was necessary, in order to get only entities measured in the gene expression data. In this case only gene and protein entities from the knowledge graph can be used. Additionally, the knowledge graph was filtered for only causal interactions, such as *increases*, *decreases*, or *regulates*, resulting in a network with 431 nodes and 673 edges. From that we only took the largest connected component to reduce the dimensionality. Hence, the used graph during our study consisted of 383 nodes and 607 edges, in which any two nodes were connected through some path.

**Clustering of filtered knowledge graph.**   One of the key aspects of iVAMBN is grouping of input features (genes, pathophysiological and clinical features) into modules in order to allow for a statistically stable identification of a Bayesian Network structure in a subsequent step. For identifying modules of genes we clustered the knowledge graph with the help of different graph clustering algorithms:

- the Markov Cluster algorithm [33, 54] implemented in the *MCL* package in R [55].

- edge betweenness [56] community detection implemented in the R package *igraph* [57]

- infomap [58] community finding method implemented in the R package *igraph* [57]

After clustering, genes being part of a single cluster were assigned to a corresponding module. Genes being not clustered but only connected to one cluster, were merged into that cluster. Genes being connected to multiple clusters were kept as single gene modules (modules consisting of a single feature) for further analysis. We selected the best clustering algorithm according

to multiple metrics described in [59] including internal density, number of edges inside clusters, average degree, expansion, cut ratio, conductance, and norm cut. Based on these metrics the average ranking of each graph clustering algorithm was computed with the rational in mind, that each cluster should have a high internal density and sparse connections across clusters. This resulted in choosing the markov clustering algorithm for further analyses. The metrics for each clustering algorithm can be found in S4 Table.

**Annotation of modules with AD disease mechanisms.**   For each module, an over-representation analysis for AD associated disease mechanisms was conducted. AD associated mechanisms were retrieved from the NeuroMMSig database [34]. For that purpose, the *enricher* function from the *clusterProfiler* package in R was used, which allows to use user-defined gene set annotations for a hypergeometric test [60]. We annotated each module with the most significant NeuroMMSig gene set after multiple testing correction via control of false discovery rate (Benjamini-Hochberg method).

## Gene expression data analysis

RNAseq data from several observational clinical studies, as well as RNAseq data from a cell line knockout experiment, were used in this work. The patient data were from i) the Religious Orders Study and Memory and Aging Project (ROSMAP) [28–30], and ii) the Mayo RNAseq Study (Mayo) [31]. The last one contains two separate datasets referring to separate brain regions, namely cerebellum and temporal cortex, while ROSMAP contains samples from the dorsolateral prefrontal cortex, head of caudate nucleus, and posterior cingulate cortex. Both studies were accessed through the AMP-AD Knowledge Portal at Synapse using the data deposited in the RNAseq Harmonization Study.

Patient samples were selected based on different criteria regarding the task they were used for:

1.  For training of the primary iVAMBN model only AD samples from the first ROSMAP batch were used, resulting in 221 samples from dorsolateral prefrontal cortex during the training phase.

2.  For external validation we used samples of the temporal cortex of AD patients in Mayo.

3.  For specificity and sensitivity analysis, samples from other batches of the ROSMAP data were used, as well as the Mayo cohort. In this step, the samples were first separated by their diagnosis, AD or healthy control, and additionally separated by their brain region, resulting in three AD and three healthy control subject subsets for ROSMAP (dorsolateral prefrontal cortex, head of caudate nucleus, and posterior cingulate cortex) and two AD subsets for Mayo (cerebellum and temporal cortex).

The Mayo study does not report Braak scores for healthy control subjects, which made us discard them from the specificity analysis, as there is no phenotype information available for them. More information about the number of samples, per brain region and study, used in each analysis step can be found in Table A in S3 Note.

The used data are gene counts provided as gene count matrices that had been generated using STAR [61]. Gene counts were normalized to log counts per Million (logCPMs) and counts from AD patients were scaled against the healthy control data within each study. That means for each AD sample and gene the corresponding mean expression value of the same gene in cognitively normal subjects was subtracted. Subsequently we divided the values by the standard deviation of the gene in healthy controls. That means raw expression values were converted into abnormality scores. For making the expression data across studies comparable,

a batch correction with ComBat [62] was applied to the scaled AD data. This normalized, scaled, and batch corrected data was then used for further analysis steps.

The cell line RNAseq data used during this study is from a THP-1 monocyte cell line with two different genetic backgrounds and two treatments. It can be found under GEO accession GSE155567. A sample could have either wild-type CD33 or a knocked out CD33 gene, plus either a control vector or a SHP-1 knock-down vector, resulting in four different conditions: i) *wild-type with control*, ii) *wild-type with SHP-1 knock-down vector*, iii) *CD33 knockout with control vector*, and iv) *CD33 knockout with SHP-1 knock-down vector*. There were 6 biological replicates per condition. Within the here presented study, only samples containing the control vector were used, resulting in twelve used samples. Therefore samples from condition 1 were called as *wild-type (WT)* samples and samples from condition 3 as *knockout (KO)* samples. Reads were aligned with STAR and gene counts were generated via the *featureCounts* function of the *Rsubread* package [63]. More details about the data can be found in [16] and under GEO accession GSE155567.

## Variational Autoencoders (VAE)

Variational autoencoders [26] are one of the most frequently used type of unsupervised neural network techniques. They can be interpreted as a special type of probabilistic graphical model, which has the form $Z \rightarrow X$, where $Z$ is a latent, usually multivariate standard Gaussian, and $X$ a multivariate random variable describing the input data. Moreover, for any sample $(x, z)$, we have $p(x \mid z) = N(\mu(z), \sigma(z))$. One of the key ideas behind VAEs is to variationally approximate

$$\log q(z|x) = \log N(z \mid \mu(x), \sigma(x)) \tag{1}$$

This means that $\mu(x)$ and $\sigma(x)$ are the multivariate mean and standard deviation of the approximate posterior $q(z \mid x)$ and are outputs of a multi-layer perceptron neural network (the encoder) that is trained to minimize for each data point $x$ the criterion

$$\log(x) \geq \frac{1}{2} \sum_{j=1}^{D} \left( 1 + \log \sigma_j(x)^2 - \mu_j(x)^2 - \sigma_j(x)^2 \right) + \frac{1}{L} \sum_l \log p(x|z^{(l)}) \tag{2}$$

Here the index $j$ runs over the $D$ dimensions of the input $x$, and $z = \mu(x) + \sigma(x) \odot \epsilon^{(l)}$ with $\epsilon^{(l)} \sim N(0, I)$ being the $l$th random sample drawn from a standard multivariate Gaussian, and $\odot$ denotes an element-wise multiplication. Notably, the right summand corresponds to the reconstruction error of data point $x$ by the model, whereas the first term imposes a regularization. We refer to [26] for more details.

## Heterogeneous Incomplete Variational Autoencoders (HI-VAE)

Variational autoencoders were originally developed for homogeneous, continuous data. However, in our case variables grouped into the phenotype module do not fulfill this assumption, because Braak stages and MMSE scores are discrete ordinal. In agreement to our earlier work [25] we thus employed the HI-VAE [35], which is an extension of variational autoencoders and allows for various heterogeneous data types, even within the same module. More specifically, the authors suggest to parameterize the decoder distribution as

$$p(x_j \mid z) = p(x_j|\gamma_j = h_j(z)) \tag{3}$$

where $h_j(\cdot)$ is a function learned by the neural network, and $\gamma_j$ accordingly models data modality specific parameters. For example, for real-valued data we have $\gamma_j = (\mu(z), \sigma_j(z)^2))$, while for ordinal discrete data we use a thermometer encoding, where the probability of each ordinal

category can be computed as

$$p(x_j = r \mid \gamma_j) = p(x_j \leq r \mid \gamma_j) - p(x_j \leq r - 1 \mid \gamma_j) \tag{4}$$

with

$$p(x_j \leq r \mid z) = \frac{1}{1 + \exp(-(\theta_j(z) - h_j(z)))} \tag{5}$$

The thresholds $\theta_j(z)$ divide the real line into $R$ regions, and $h_j(z)$ indicates, in which region $z$ falls. The data modality specific parameters are thus $\gamma_j = \{h_j(z), \theta_1(z), \ldots, \theta_{R-1}(z)\}$ and are modeled as output of a feed forward neural network.

According to [35] we use batch normalization to account for differences in numerical ranges between different data modalities.

For multi-modal data and in particular discrete data a single Gaussian distribution may not be a sufficiently rich representation in latent space. Hence, the authors propose to replace the standard Gaussian prior distribution imposed for $z$ in VAEs by a Gaussian mixture prior with $K$ components:

$$s \sim Categorical(\pi) \tag{6}$$

$$z \mid s \sim N(\mu(s), I_K) \tag{7}$$

where $\pi_k = \frac{1}{K}$ for $k = 1, 2, \ldots, K$ and $s$ is a one-hot vector encoding of the mixture component. We evaluated different choices of $K$ using a 3-fold cross-validation, while employing the reconstruction error $\frac{1}{L} \sum_l \log p(x|z^{(l)})$ as an objective. In conclusion it turned out that $K = 1$ component was an optimal choice for all modules in our iVAMBN model.

## Modular bayesian networks

Let $X = (X_v)_{v \in V}$ be a set of random variables indexed by nodes $V$ in a directed acyclic graph (DAG) $G = (V, E)$. In our case each of these nodes corresponds either to lower dimensional embedding of a group of variables (i.e. module) in the original data, or to an original features (e.g. biological sex) in the dataset. According to the definition of a Bayesian Network (BN), the joint distribution $p(X_1, X_2, \ldots, X_n)$ factorizes according to

$$p(X_1, X_2, \ldots, X_n) = \prod_{v \in V} p(X_v \mid X_{pa(v)}) \tag{8}$$

where $pa(v)$ denotes the parent set of node $v$ [27]. In our case random variables follow either a Gaussian or a multinomial distribution, i.e. the BN is hybrid. Notably, no discrete random variable was allowed to be a child of a Gaussian one.

Since the BN in our case is defined over low dimensional representations of groups of variables, we call the structure Modular Bayesian Network (MBN). Notably, a MBN is a special instance of a hierarchical BN over a structured input domain [64–67].

A typical assumption in (M)BNs is that the set of parameters $(\theta_v)_{v \in V}$ associated to nodes $V$ are statistically independent. For a Gaussian node $v$ parameters can thus be estimated by fitting a linear regression function with parents of $v$ being predictor variables [27]. Similarly, for a discrete node $\tilde{v}$ having only discrete parents, parameters can be estimated by counting relative frequencies of variable configurations, resulting into a conditional probability table.

## Quantitative modeling across biological scales via iVAMBN

**Model training.** The here presented *Integrative Variational Autoencoder Modular Bayesian Network (iVAMBN)* approach (Fig 1), integrates different biological scales together with a knowledge graph into the previously published Variational Autoencoder Modular Bayesian Network (VAMBN) approach [25]. More precisely, there are four steps to build an iVAMBN model: i) Definition of modules of variables, ii) Training of a HI-VAE for each module, iii) Definition of logical constraints for possible edges in the MBN, and iv) Structure and parameter learning of the MBN using encoded values for each module. These four steps result from the fact that HI-VAEs (as well as any other variants of variational autoencoders) themselves can be interpreted as specific types of BNs and thus the overall log-likelihood of an iVAMBN model can be decomposed accordingly. That means the overall iVAMBN model can be interpreted as a special type of Bayesian Network, see [25] for details.

The four model building steps were followed in the application of the iVAMBN approach in this work as well. The modules of variables were mainly defined through the previously explained Markov clustering of the knowledge graph, plus an additional module summarizing MMSE (Mini–Mental State Examination) and Braak stages into one *phenotype* module. MMSE measures cognitive impairment by testing the orientation in time and space, recall, language, and attention, while Braak stages refer to the degree of biological brain pathology [68]. Some non-assigned genes, were directly treated as nodes in the MBN construction and thus also called gene modules. The same was done for demographic features, like sex, age, years of education and the APOE genotype.

For training the HI-VAEs for each module a hyperparameter optimization (grid search) was implemented over learning rate (learning rate $\in \{0.001, 0.01\}$) and minibatch size (minibatch size $\in \{16, 32\}$) as in [25]. Each parameter combination was evaluated with the reconstruction loss as objective function in a 3-fold cross-validation scenario.

In general the number of possible MBN DAG structures for $n$ nodes grows super-exponentially with $n$ [24], making identification of the true graph structure highly challenging. Therefore, our aim was to restrict the set of possible DAGs a priori as much as possible via knowledge based logical constraints. More specifically we imposed the following causal restrictions:

- Nodes defined by demographic or clinical features (like age, gender, APOE genotype, and brain region) can only have outgoing edges.

- The phenotype module (= clinical outcome measures) can only have incoming edges.

- Genes and gene modules can not influence demographic or clinical features, except the age.

To additionally integrate prior knowledge defined through the knowledge graph, we tested three different strategies while building a MBN:

1. **Completely data driven**: The knowledge graph is completely ignored for structure learning.

2. **Knowledge informed**: The knowledge graph is used in the greedy hill climbing algorithm for structure learning i) as starting point, ii) as white list (intending that those edges were defined as pre-existing), or iii) as both.

3. **Completely knowledge driven**: The knowledge graph provides the structure of the MBN and additional connections are only allowed for demographics or the phenotype module.

Structure learning of the MBN was always performed via a greedy hill climber using the Bayesian Information Criterion for model selection. We employed the implementation provided in R-package *bnlearn* [69].

**Evaluating the model fit.**  To evaluate the fit of the overall iVAMBN model we employed the generative nature of our model: Following a topological sorting of the nodes of the DAG of the MBN we first sampled from the distribution of each node conditional on its parent. Notably, for MBN nodes representing modules this amounted to sample from the posterior of the according HI-VAE, which in practice can be realized via injection of normally distributed noise, see Section Variational Autoencoders, Eq (2). Subsequently, the random sample was then decoded via the HI-VAE. Altogether we thus generated as many synthetic subjects as real ones. We then compared the marginal distribution of each variable based on the synthetic and the real data. Results, including summary statistics and Kullback-Leibler divergences are shown in the supplementary material (Fig A and B in S4 Note). Furthermore, we compared the correlation matrices of synthetic and real data.

## CD33 down-expression simulation and analysis

To be able to simulate a down-expression of CD33, we first shifted the distribution of CD33 such that it reflects a 9-fold down-expression of CD33. In agreement to the theory of Bayesian Networks this operation made CD33 conditionally independent of its parents in the MBN, which amounts to deleting any of its incoming edges and resulted into a mutilated MBN. Afterwards we exploited the fact that iVAMBN is a generative model. That means we first drew samples from the conditional densities of the mutilated MBN. Practically this amounted to first topologically sort the nodes in the MBN, hence exploiting the fact that the underlying graph structure cannot have cycles. Subsequently, samples were drawn from the statistical distribution of each node while conditioning on the value of its parents. The result was a per-sample module activity score, which we then decoded through our HI-VAE models into single gene scores.

Differences between the wild-type and simulated down-expression samples were investigated afterwards via multiple statistical hypothesis tests: First, a linear regression was used to model the down-expression effect on gene expression and on the different phenotype scores. Second, the *globaltest* package in R was used to test the differential expression of specific gene sets between the wild-type and simulated down-expression group [49]. Those tested gene sets were here defined through the modules' genes used in the MBN, meaning that we tested for differential expression of MBN's gene modules. P-values were adjusted for multiple test scenario with the help of the *subsets* option of *globaltest* and via calculating the false discovery rate. The globaltest for gene sets, as well as the fold change analysis, was also applied to the cell line WT and KO data to be able to validate the results.

Effects of the perturbation of other candidate targets were simulated similarly as the CD33 knock-down. Again, the distribution of the respective target was shifted such that it reflected a 9-fold down- or up-regulation. The module was identified to which the candidate target had been assigned, and all variables (including the perturbed target) mapping to that module were encoded via the previously trained HI-VAE for the module. Subsequently, the effects on the phenotype could be predicted in the same way as described for CD33.

## Supporting information

**S1 Table. Module enrichment analysis.** If the genes in a module do not enrich NeuroMMSig terms significantly (adjusted $p < 0.05$), individual genes are reported. If significant enriched terms could be found, all significant pathways are reported.
(XLSX)

**S2 Table. Module assignment.** For every gene the corresponding module number from the Markov clustering is given. Module 0 refers to all standalone genes.
(XLSX)

**S3 Table. Bootstrap confidence results.** This is the full list of the bootstrap confidence of each possible edge in the Bayesian Network. For every edge the corresponding start and end note, as well as, the bootstrap strength and the direction is given.
(XLSX)

**S4 Table. Graph Clustering Metrics.** The three clustering algorithms: 1) markov clustering, 2) edge betweenness, and 3) infomap were applied on the knowledge graph. For every cluster algorithm the corresponding metrics are given, as well as, the average rank. Printed in bold is the best algorithm according to the respective metric. The algorithms were ranked per metric and the average rank per algorithm was calculated.
(XLSX)

**S1 Note. iVAMBNs Module Definition.**
(PDF)

**S2 Note. iVAMBNs Knowledge Integration.**
(PDF)

**S3 Note. Specificity and sensitivity analysis.**
(PDF)

**S4 Note. Evaluating the model fit.**
(PDF)

**S1 Fig. Clustered knowledge graph.** Knowledge graph modules (clusters) are annotated with significantly enriched (adjusted $p < 0.05$) NeuroMMSig mechanisms. If the genes in a module do not enrich NeuroMMSig terms significantly, symbols of contained genes are reported. If multiple significant enriched terms could be found, the most significant pathway was used for naming the corresponding node. In case that a module contains a single gene, the gene symbol is reported. CD33 is marked in red, while other single genes are displayed in green, and non-single gene modules in purple.
(PNG)

**S2 Fig. Quantitative effect between modules of shortest path.** Each correlation (R) is shown along with its confidence interval (CI) and multiple testing adjusted p-value. Left: Correlation of CD33 with prostaglandin pathway module. Right: Correlation of prostaglandin pathway module with the phenotype module.
(PNG)

**S3 Fig. Quantitative effect between modules of newly trained edges with confidence 1.** Each correlation (R) is shown along with its confidence interval (CI) and multiple testing adjusted p-value. The *from* module is always shown on x-axis while the *to* module is shown on y-axis.
(PNG)

**S4 Fig. Overlap of ROSMAP and Mayo network structures.** The overlap of the independent bootstrap structure learning for ROSMAP data and Mayo data is shown for different threshold values. The black line represents the overlap when considering the direction of the edge, the dashed line the overlap of the network skeletons.
(PNG)

**S5 Fig. Effects on phenotype scores of up- and down-regulation simulations.** The bar plots show the difference between the mean score in the original data and the mean score in the simulated data for each target and each phenotype score, namely MMSE (upper two rows) and Braak score (bottom two rows). First and third row shows the results of under-expression, while second and forth rows shows the results of over-expression.
(PNG)

## Acknowledgments

The results published here are in whole or in part based on data obtained from the AD Knowledge Portal (https://adknowledgeportal.org). Data generation was supported by the following NIH grants: P30AG10161, P30AG72975, R01AG15819, R01AG17917, R01AG036836, U01AG46152, U01AG61356, U01AG046139, P50 AG016574, R01 AG032990, U01AG046139, R01AG018023, U01AG006576, U01AG006786, R01AG025711, R01AG017216, R01AG003949, R01NS080820, U24NS072026, P30AG19610, U01AG046170, RF1AG057440, and U24AG061340, and the Cure PSP, Mayo and Michael J Fox foundations, Arizona Department of Health Services and the Arizona Biomedical Research Commission. We thank the participants of the Religious Order Study and Memory and Aging projects for the generous donation, the Sun Health Research Institute Brain and Body Donation Program, the Mayo Clinic Brain Bank, and the Mount Sinai/JJ Peters VA Medical Center NIH Brain and Tissue Repository. Data and analysis contributing investigators include Nilüfer Ertekin-Taner, Steven Younkin (Mayo Clinic, Jacksonville, FL), Todd Golde (University of Florida), Nathan Price (Institute for Systems Biology), David Bennett, Christopher Gaiteri (Rush University), Philip De Jager (Columbia University), Bin Zhang, Eric Schadt, Michelle Ehrlich, Vahram Haroutunian, Sam Gandy (Icahn School of Medicine at Mount Sinai), Koichi Iijima (National Center for Geriatrics and Gerontology, Japan), Scott Noggle (New York Stem Cell Foundation), Lara Mangravite (Sage Bionetworks).

## Author Contributions

**Data curation:** Tamara Raschka, Bruce Schultz, Christian Ebeling.

**Formal analysis:** Tamara Raschka, Meemansa Sood.

**Funding acquisition:** Holger Fröhlich.

**Methodology:** Tamara Raschka, Meemansa Sood, Bruce Schultz, Aybuge Altay, Christian Ebeling, Holger Fröhlich.

**Software:** Tamara Raschka, Meemansa Sood, Bruce Schultz, Christian Ebeling.

**Supervision:** Holger Fröhlich.

**Visualization:** Tamara Raschka.

**Writing – original draft:** Tamara Raschka, Holger Fröhlich.

**Writing – review & editing:** Holger Fröhlich.

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
