## [Decision Letter · Decision Letter 0]

13 Apr 2022

Dear Mrs Raschka,

Thank you very much for submitting your manuscript "AI reveals insights into link between CD33 and cognitive impairment in Alzheimer's Disease" for consideration at PLOS Computational Biology.

As with all papers reviewed by the journal, your manuscript was reviewed by members of the editorial board and by several independent reviewers. In light of the reviews (below this email), we would like to invite the resubmission of a significantly-revised version that takes into account the reviewers' comments.

Reviewer 1, in particular, raises a lot of valid points that need to be addressed. Specifically on the topic of clustering, I agree that comparing to more than just one method would be appropriate, although I do not share the reviewer's viewpoint of MCL being one of the worse algorithms, but you need to address all the reviewers points.

We cannot make any decision about publication until we have seen the revised manuscript and your response to the reviewers' comments. Your revised manuscript is also likely to be sent to reviewers for further evaluation.

Sincerely,

Lars Juhl Jensen

Associate Editor

PLOS Computational Biology

Arne Elofsson

Deputy Editor

PLOS Computational Biology

Reviewer's Responses to Questions

**Comments to the Authors:**

Reviewer #1: I'm enthusiastic about your goal of doing in silico screening of the effects of perturbing specific genes, and I think you've assembled a nice array of technologies for doing that. However, there are limitations of the implementation, and more to the point, after reading this several times I really couldn't tell what you were doing on several points - the take away from this is that if someone who cares way more about figuring out your methods than a casual read does, there are indeed issues. I've tried to make specific questions and points suggestions about this.

Need refs for sentences 1&2.

I'm sure you know what you did, but the use of "module" and "cluster" in multiple contexts or with distinct contents can be (is) confusing when you don't clearly note what type of cluster of module you're referring to. Deploying a simple descriptor, as in "we used these [gene] modules in tandem with the clusters [of proteins] in order to....." where you would insert the modifier in []. This situation comes up so many times, so please examine every usage of those words and clarify.

Right around line 108 there should be a statement of why you're deploying VAMBN, other than just you have it on hand.

How specific are these results to cd33? Do you get similar ones if you put other genes in the microglia module through this?

Is the approach fast enough that you can use it for screening? i.e. put every gene through it to prioritize disease molecular systems

What is the false positive rate associated with table 3?

Considering the significance calculation of the similarity of the bayesian networks from the two cohorts - wonder if that's realistic as they have the same underlying the same knowledge graph, so is it an independent replication?

There are single genes listed in table 3 - is that really a "mechanism"?

You're being pretty restrictive with AD if you only end up with 221 samples in rosmap - I'd want to know exactly what your criteria was for that - there are ~1600 dlpfc samples available and we show that most brains have some level of AD pathology. Do you know conclusively that the AD graph is indeed somehow specific to AD brains? Why not run it on some definition of healthy and see how it differs? You could also get data from at least three other regions in ROSMAP with several hundred gene expression arrays - would be a big boost to validating your model.

How does the data reduction in the inner layers of the autoencoder compare to the conditional dependencies of bayesian networks?

Why don't you incorporate the score of the networks in addition to edge frequency?

What is your edge updating function for the bayes net?

Not clear if you're training an encoder per each cluster of genes in the knowledge graph or treating each cluster as a single variable, and then the autoencoder handles this group of meta variables.

Should provide some guidance on the primacy of gene expr vs knowledge graph and how that stems from structure of algo

When you say AD-specific mechanisms L126, do you really mean it? i.e. if a gene was ever implicated in some other disease you'd exclude it?

Need to set up the integration of knowledge graphs and expression with some context - ideally data driven. i.e. what is the average coexpression value for various knowledge graphs? Are we combining two data sources with some similarity or no similarity greater than chance?

Why did you chose markov clustering for the K-graph? That's basically the method that always loses in most clustering methods comparisons:) I'd want to see at least one more clustering method applied and equivalent results coming out, as well as some notes on the modularity of the partition you find.

Not defined in fig 1 if knowledge graph has green and blue balls denoting genes or if those are modules of genes. What are BL and WL?

l133 "here: gene clusters and phenotype related scores," do you mean that there are genes AND phenos in the same module or they are each in their own module?

Figure 1 caption "constraining potential connections between them" do you mean only connections between groups of variables or between pairs of variables within a group or both?

Figure 1 design needs to be revamped. You know what you did, and I'm sure the fig is one accurate depiction of that, but if you don't know what you did, it's not sufficient to figure out what you did.

Reviewer #2: Summary:

An interesting approach to understanding mechanisms of Alzheimer's disease that blends AI-based data-driven feature generation with knowledge graphs. Experiments included both human and mouse data.

Details:

Apologies for the brevity of this review. Time is precious.

Conceptually the method is sound and the implementation as described is appropriate and understandable. Briefly, input features (genetics, biomarkers, clinical) are mapped into modules using clustering plus some manual editing. A VAE is used for dimensionality reduction of the modules into a Gaussian or multinomial parametrization, which become the nodes of a BayesNet, with network parameter learning constrained by a knowlege graph and a few sensible causality constraints.

Cross-cohort harmonization was used to adjust for batch effects in the human data.

The method is an intelligent augmentation of the authors' existing VAMBN to incorporate existing knowledge into data-driven modelling.

The authors make grand/general statements about their model (don't we all...) and the possible ramifications for identifying drug targets, but demonstrate only one result on the possible effects of CD33 downregulation. I suspect that this was likely due to the availabiltiy of mouse data for validation, but I think the reader would benefit from an additional results section where the authors analyse other potential targets, and discuss them in the context of the literature. In particular, it would be valuable to consider two categories of targets: 1) well-studied therapeutic targets such as those related to existing GWAS studies and/or clinical trials (e.g., of anti-amyloid and anti-tau drugs); 2) potential novel targets. This would add considerable value to this study, and generate much discussion in the field.

I appreciated the method's aim to circumvent the need for fitting a differential equation model. But I wonder if there is any scope for using longitudinal data to validate the model — if such data exists, of course.

= Comments =

Methods: It makes sense to retrain the ROSMAP model for validation on the Mayo dataset. What is not clear from the manuscript is the consistency of the two trained models (with and without MMSE) and analysis of any differences. Suggest that the authors provide results of an analysis that compares the two ROSMAP-trained models, e.g., edge strength differences (if any).

Results: CD33 down-expression simulation results are interesting. The discussion around caveats (starting on line 311) raises questions. Primarily, the potential safety issues - have the authors considered including such issues as additional constraints in the model? For example, explicitly excluding unsafe targets for therapy from the knowledge graph seems sensible (but may have implications I haven't thought of). I (and the readers of this manuscript) would be very interested in the authors' thoughts on the matter.

Further comments and suggestions are below.

- Line 160: A github repository can be deleted, so the full list of molecules within each cluster should be included in Supplementary Material.

- Line 193: Please explain the rationale behind your choice to use 40% as the threshold.

- Line 216, Supplementary Figure S3: A linear association is surely not suitable for MMSE. From the data it is apparent that the MMSE ceiling is undersampled, but the floor appears to be well sampled. What about fitting a sigmoid instead?

- Figure 5: A typo in the caption: I think "results in higher scores" should be "results in lower scores".

- Throughout: remove "highly significant". Statistical significance thresholds are arbitrary.

- Throughout: give it another proof reading, some typos exist.

**Have the authors made all data and (if applicable) computational code underlying the findings in their manuscript fully available?**

Reviewer #1: Yes

Reviewer #2: Yes

PLOS authors have the option to publish the peer review history of their article (what does this mean?). If published, this will include your full peer review and any attached files.

Reviewer #1: No

Reviewer #2: No
---

## [Decision Letter · Decision Letter 1]

22 Jun 2022

Dear Mrs Raschka,

Thank you very much for submitting your manuscript "AI reveals insights into link between CD33 and cognitive impairment in Alzheimer's Disease" for consideration at PLOS Computational Biology.

As with all papers reviewed by the journal, your manuscript was reviewed by members of the editorial board and by one of the reviewers from the previous round. In light of the review (below this email), we would like to invite the resubmission of a significantly-revised version that takes into account the reviewers' comments.

Thanks for considerably improving the manuscript compared to the previous round. However, one of the reviewers still has a couple of comments that need to be addressed as well as some suggestions for how to improve the broader impact of the work, which I recommend that you at least consider.

We cannot make any decision about publication until we have seen the revised manuscript and your response to the reviewers' comments. Your revised manuscript is also likely to be sent to reviewers for further evaluation.

Sincerely,

Lars Juhl Jensen

Associate Editor

PLOS Computational Biology

Arne Elofsson

Deputy Editor

PLOS Computational Biology

Reviewer's Responses to Questions

**Comments to the Authors:**

Reviewer #1: Dear authors,

I appreciate the authors responses to these critiques, and I think the paper is resultantly clearer and will reach a broader audience.

A few large-scale issues remain:

Network generation:

There's a relatively easy way to increase the accuracy of your Bayesian networks, which would be to use REV-move or some other more advanced edge updating rule, because as you're aware, it's very easy for structure-move to become stuck, particularly in such large networks.  There are implementations of this tested on gene expression data that showed several times the accuracy of structure-move, so this is a relatively easy way to drastically improve accuracy - particularly important, again, in light of the easy with which struct-move gets stuck.

Validation:

The Mayo validation is helpful.  However, there are other datasets which are actually larger (n>1000) and from other brain regions that will allow you to much more accurately estimate networks.  Just go to the RADC website and request gene expression data - it's all freely available.  For modern omics studies I would expect to see such tests ,as the data is easily available, and it will go a long way to convincing readers of the reliability of the method.

Model components:

One of the most intriguing aspects of this work is the integration of prior knowledge in the form of a graph.  Opinions about the validity of such knowledge sources vary, so it would be helpful to see to what extent this helps to increase predictive accuracy.  Ideally, the knowledge graph would be completely ablated and then predictive accuracy reassessed in all aspects.  If that is not possible, it should be possible to, say, delete 25, 50 and 75% of the edges in the graph and rerun to assess its contributions.  Similarly, a varying number of incorrect edges could be added to the graph.  This would provide an estimate of the sensitivity of the knowledge graph to new information, as well as it's total contribution to predictions.

Screening:

I think many users of your work will be interested to sift through the membership of a gene list, based on their importance, as determined by your method.  This seems to entail just putting a for loop around your method, grabbing the knowledge graph from ref 10, or a sub graph, and rerunning.  As motivation, I've had multiple pharma specifically ask me not just to reiterate known targets (i.e. cd33), but want info on what a given computational method has predicted that is not already popularly supported.  In summary, I think it would be very useful to share the screening results for all microglia genes - it would demonstrate this is feasible and provide directly useful results.

Minor:

For the cd33 KO data, if you look at the genes which did not match your prediction, do they share some common feature?  This might enable you to comment on some biological process which is outside of the scope of the method (possibly for understandable reasons).

**Have the authors made all data and (if applicable) computational code underlying the findings in their manuscript fully available?**

Reviewer #1: Yes

PLOS authors have the option to publish the peer review history of their article (what does this mean?). If published, this will include your full peer review and any attached files.

Reviewer #1: No
---

## [Decision Letter · Decision Letter 2]

3 Sep 2022

Dear Mrs Raschka,

Thank you very much for submitting your manuscript "AI reveals insights into link between CD33 and cognitive impairment in Alzheimer's Disease" for consideration at PLOS Computational Biology.

As with all papers reviewed by the journal, your manuscript was reviewed by members of the editorial board and by several independent reviewers. In light of the reviews (below this email), we would like to invite the resubmission of a significantly-revised version that takes into account the reviewer's remaining comments.

Having looked at the comments made by the reviewer last round and in this round, I agree with the reviewer that several of the points already raised before have not yet been addressed, and that this needs to be done for the manuscript to be accepted.

We cannot make any decision about publication until we have seen the revised manuscript and your response to the reviewers' comments. Your revised manuscript is also likely to be sent to reviewers for further evaluation.

Sincerely,

Lars Juhl Jensen

Academic Editor

PLOS Computational Biology

Arne Elofsson

Section Editor

PLOS Computational Biology

Reviewer's Responses to Questions

**Comments to the Authors:**

Reviewer #1: Thank you for your responses and additions to the text to date.  Below are the major remaining issues.

For the knowledge graph, what if you use something else (louvain or other mainstream options) other than markov clustering (which does not fare well in comparisons of clustering methods) - how much do the modules change and how much do those changes impact performance.  Would anticipate that the exact choice of algorithm shouldn't affect results, but if it does, would be important to note this sensitivity.

As the model has several major components, helpful to know to what extent each contributes to the prediction.  For instance, if you scramble the edges in the knowledge graph, cluster that and use it, how much does accuracy fall?

The titular claim is that this is a network for Alzheimer's disease.  You restrict your subjects down to 200, but by using all the DLPFC data you would have 1200+samples, and it's not been proven that the resulting network from that would be different from this one.  Indeed, many cognitively healthy subjects will have brain pathology.  You can either try to prove that the network is AD-specific, or just include a broader range of samples to increase the accuracy of the findings or perform cross-validation.  In either case, you will need to explore the full expression data set.

I am puzzled by the statement that "brain expression is known to be region specific" as a reason not to repeat this analysis with data from other regions - if anything, that is a reason TO test it in other regions.  In any case, it's very standard to repeat tests in all regions - even if you think it will not be replicated - in order to see how the results vary across regions.  You can see that several hundred samples are available in ROSMAP for PCC and AC, as well as ~200 in the hypothalamus and nucleus accumbens.  Furthermore there are 1200+dlpfc samples available as mentioned previously.  Testing in all of these data are essential to measuring the robustness of the findings.  I've asked that these be made more obvious on the Rush resource page, but in any case just enter a limited data request DUA and in text say you're requesting gene expression data for all of these regions and you'll receive it.

**Have the authors made all data and (if applicable) computational code underlying the findings in their manuscript fully available?**

Reviewer #1: Yes

PLOS authors have the option to publish the peer review history of their article (what does this mean?). If published, this will include your full peer review and any attached files.

Reviewer #1: No
---

## [Editor Report · Decision Letter 3]

18 Jan 2023

Dear Mrs Raschka,

We are pleased to inform you that your manuscript 'AI reveals insights into link between CD33 and cognitive impairment in Alzheimer's Disease' has been provisionally accepted for publication in PLOS Computational Biology. We apologize for the long time the last round of revision took — we tried to get one last round of comments from the reviewer, but this proved impossible.

Best regards,

Lars Juhl Jensen

Academic Editor

PLOS Computational Biology

Arne Elofsson

Section Editor

PLOS Computational Biology

---

## [Editor Report · Acceptance letter]

6 Feb 2023

PCOMPBIOL-D-22-00173R3 

AI reveals insights into link between CD33 and cognitive impairment in Alzheimer's Disease

Dear Dr Raschka,

I am pleased to inform you that your manuscript has been formally accepted for publication in PLOS Computational Biology. Your manuscript is now with our production department and you will be notified of the publication date in due course.

With kind regards,

Anita Estes
